# Successful Treatment of Acute Uric Acid Nephropathy with Rasburicase in a Primary Central Nervous System Lymphoma Patient Showing a Dramatic Response to Methotrexate—Case Report

**DOI:** 10.3390/jcm11195548

**Published:** 2022-09-22

**Authors:** Yoshihiro Mouri, Manabu Natsumeda, Noritaka Okubo, Taro Sato, Taiki Saito, Kohei Shibuya, Shiori Yamada, Jotaro On, Yoshihiro Tsukamoto, Masayasu Okada, Makoto Oishi, Takeyoshi Eda, Junko Murai, Hiroshi Shimizu, Akiyoshi Kakita, Yukihiko Fujii

**Affiliations:** 1Department of Neurosurgery, Brain Research Institute, Niigata University, Niigata 951-8585, Japan; 2Department of Pathology, Brain Research Institute, Niigata University, Niigata 951-8585, Japan; 3Division of Pharmacy, Medical and Dental Hospital, Niigata University, Niigata 951-8520, Japan; 4Institute for Advanced Biosciences, Keio University, Tsuruoka 997-0052, Japan

**Keywords:** PCNSLs, high-dose methotrexate, acute uric acid nephropathy, rasburicase

## Abstract

Background: Primary central nervous system lymphomas (PCNSLs) are sensitive to chemotherapy. The standard treatment is high-dose methotrexate (MTX)-based chemotherapy. There are no reports of successful treatment of acute uric acid nephropathy with rasburicase after MTX administration in PCNSLs. Case presentation: A 54-year-old man with a history of gout presented with a change in character and cognitive dysfunction. MRI showed a large enhancing mass spanning the bilateral frontal lobes and the right temporal lobe. After endoscopic biopsy, an MTX, procarbazine, and vincristine (MPV) regimen was initiated for the treatment of the PCNSL. After the initiation of chemotherapy, the patient experienced a gout attack, and blood examination revealed acute renal failure (ARF) and hyperuricemia. The considered causes of ARF included MTX toxicity and acute uric acid nephropathy. As the dramatic effect of MTX was observed, treatment was continued despite ARF, most probably due to acute hyperuricemia due to tumor lysis, which was treated in parallel. After an improvement in renal function, MTX was resumed, and rasburicase was initiated to control hyperuricemia. A complete response was obtained after induction chemotherapy. Hyperuricemia was controlled with rasburicase, and renal function was preserved. Conclusions: Acute uric acid nephropathy should be considered when ARF occurs after the initiation of MTX in PCNSLs, especially in newly diagnosed PCNSL patients with large tumors or hyperuricemia.

## 1. Introduction

Primary central nervous system lymphomas (PCNSLs) account for approximately 4% of all primary brain tumors [1]. Sensitivity to high-dose methotrexate (HD-MTX) has been observed in multiple prospective and retrospective studies, and HD-MTX-based regimens are standard for induction chemotherapy [2,3]. As PCNSLs are sensitive to chemotherapy, acute uric acid nephropathy due to tumor lysis can occur.

Rasburicase is a recombinant version of uric acid oxidase, an enzyme that metabolizes uric acid to allantoin, and is a highly effective treatment for hyperuricemia during chemotherapy. In the present case, we report for the first time the successful control of hyperuricemia by rasburicase after MTX-induced acute uric acid nephropathy in PCNSLs.

## 2. Case Presentation

A 54-year-old Japanese man presented with a change in character and memory loss beginning one month prior to the initial presentation. The patient had a history of gout and hyperuricemia, which were well-controlled without medication prior to presentation. Magnetic resonance (MR) imaging showed a large lesion spanning both frontal lobes, the right temporal lobe, and the corpus callosum. The lesion showed hypointensity on T1-weighted imaging (T1WI) (Figure 1A), hyperintensity on T2WI (Figure 1B), fluid-attenuated inversion recovery (FLAIR) (Figure 1C) and diffusion-weighted imaging (DWI) (Figure 1D), homogeneously enhanced on post-contrast MR images (Figure 1E). A pronounced perifocal edema was observed. The lesion was characterized by slightly high density on pre-contrast computed tomography (CT) with accompanying perifocal low density, and homogeneous enhancement was observed on post-contrast CT (Figure 1F). No systemic lesions were detected on pre- or post-contrast body CT scans.

The lesion in the right frontal lobe was endoscopically biopsied. Histological examination revealed perivascular and diffuse distribution of relatively large lymphoma cells (Figure 2A,B) positive for CD20 (Figure 2C) and smaller lymphocytes thought to be reactive. The tumor cells were highly positive for SLFN11, a marker of sensitivity to DNA-damaging agents (Figure 2D).

Intravenous dexamethasone was started at a dose of 16 mg/day and tapered. Subsequently, the tumor size considerably decreased, and the symptoms improved before the commencement of chemotherapy (Figure 3A,B).

For induction therapy, MTX (3.5 g/m^2^), procarbazine (100 mg/m^2^/d for 7 days), and vincristine (1.4 mg/m^2^) (MPV regimen) were simultaneously started (Day 0) in accordance with our previously published protocol [4]. After administration, the patient experienced nausea, appetite loss, diarrhea, and acute swelling, discoloration, and severe pain in the left ankle. Before the initiation of induction chemotherapy, no abnormal data were observed in the complete blood count or biochemical testing. Blood examination revealed acute renal failure (serum Cre, 4.87 mg/dL; BUN, 31 mg/dL; eGFR, 10.93 mL/min/1.73 m^2^) on Day 2 and hyperuricemia (serum UA, 10.5 mg/dL) on Day 6. Urine volume was maintained at 2520–2880 mL/d and electrolytes, including serum potassium, phosphorus, and calcium, were within the normal range. Therefore, ARF was treated by increasing the volume of intravenous fluids and the dosage of leucovorin, MTX was eliminated from the plasma after 10 days [5], and the renal function was normalized over a period of 3 weeks (Figure 4). Diuretics and blood dialysis were not used.

ARF was thought to be due to either acute MTX toxicity or hyperuricemia. Because clinical and radiographical response to treatment was observed, we decided to continue MTX treatment by decreasing the dose of the second MTX course to 1.5 mg/m^2^ and treating hyperuricemia with rasburicase; 0.2 mg/kg rasburicase was administered from Day 28 to Day 31, and 1.5 mg/m^2^ of MTX on Day 29. Serum uric acid levels did not increase after the second course of MTX, although swelling of the right ankle was observed, and the renal function was preserved. Similarly, rasburicase was administered from Day 42 to Day 45, and the third course of MTX was administered at a maximal dose of 3.5 mg/m^2^ on Day 42 without complications. The fourth and fifth courses of MTX (3.5 mg/m^2^) were administered on Days 57 and 71, respectively, without complications. The Naranjo Adverse Drug Reaction Probability Scale score was 11 out of a possible 13, suggesting a definite probability that MTX was responsible for ARF and hyperuricemia. Due to concerns regarding a relapse of renal dysfunction after continuation of MTX, hyperuricemia was treated with febuxostat and not allopurinol. On Day 80, Ara-C (2000 mg/m^2^/d for two days) was successfully administered, and the patient was discharged. Post-contrast MR images showed a dramatic response (unconfirmed complete response (CRu)) after three courses (Figure 3C) of MTX and a complete response (CR) after five courses (Figure 3D). Dramatic improvements in the cognitive function were observed after induction chemotherapy (the Mini-Mental State Examination (MMSE) improved from 21/30 to 28/30 and the Wechsler Adult Intelligence Scale 4th edition (WAIS-IV) total IQ improved from 54 to 98), and the patient was able to return to work.

Informed written consent was obtained from the patient for the publication of this case report and inclusion of clinical and imaging details in the manuscript.

## 3. Discussion

Tumor lysis syndrome (TLS) is an oncological emergency in which metabolic abnormalities, such as hyperkalemia, hyperphosphatemia, hypocalcemia, and hyperuricemia, occur as a result of a large number of tumor cells being killed by treatment. Hyperuricemia occurs because of the rapid metabolism of nucleic acids. There are two types of TLS: laboratory TLS (LTLS) and clinical TLS (CTLS). LTLS is defined by the presence of two or more of the following criteria: hyperuricemia, hyperkalemia, hyperphosphatemia, and/or hypocalcemia between the 3 days before treatment and the 7 days after treatment. CTLS is defined by one of the following: renal dysfunction of more than 1.5 times of the upper limit of normal (ULN), arrhythmia/sudden death, or convulsions in addition to LTLS [6]. The suggested risk factors for TLS include tumor-related factors, such as large tumor volume, rapid growth rate, and high sensitivity to chemotherapy, and patient- or history-related factors, such as the presence of hyperuricemia or hyperphosphatemia before chemotherapy, history of nephropathy or prior exposure to nephrotoxic agents, oliguria, aciduria, and dehydration [6]. An expert panel suggested that adult systemic diffuse large B cell lymphomas with elevated LDH levels and bulky solid tumors are of a high risk for TLS [7].

The present case does not fit the definition of LTLS because of a lack of hyperkalemia, hyperphosphatemia, and/or hypocalcemia; however, after the initiation of induction therapy, hyperuricemia and ARF were observed. MR images taken before the treatment showed a large tumor volume, which was dramatically reduced after steroid and MTX treatment. In addition, the patient had a history of hyperuricemia and experienced symptoms of loss of appetite, nausea, and diarrhea, probably owing to the adverse effects of induction chemotherapy, leading to relative dehydration despite ample hydration. Therefore, this patient should have been regarded as having a high risk of TLS. We hypothesized that ARF was caused by acute uric acid nephropathy due to a TLS-like pathophysiology. Renal toxicity due to MTX was also considered. However, since the renal function was preserved after repeated MTX administration, we believed that the former was more likely.

Acute uric acid nephropathy is caused by the deposition of uric acid in the distal convoluted tubules and the collecting duct. Allopurinol, febuxostat, and rasburicase are known to lower serum uric acid levels [8,9]. Allopurinol is a hypoxanthine analog known to inhibit xanthine oxidase, which oxidizes xanthine and hypoxanthine, leading to uric acid production. Febuxostat is a selective xanthine oxidase inhibitor. Lastly, rasburicase is a recombinant version of uric acid oxidase, an enzyme that metabolizes uric acid to allantoin and is not found in humans [6]. Rasburicase is superior to allopurinol in lowering serum uric acid levels during MTX treatment in acute lymphoblastic leukemia [8]. Febuxostat has biliary elimination and does not require dose adjustment in patients with renal impairment [10] and has been shown in a meta-analysis to be as effective as allopurinol in TLS prophylaxis, although allopurinol is still the mainstay [11]. In the present case, we administered rasburicase before and during the second and third courses of MTX treatment in addition to febuxostat. Uric acid levels did not increase, and the renal function was preserved during the induction and consolidation treatments. The MR images taken after the third course of MTX showed that the tumor volume had dramatically decreased, and, because of the risk of producing anti-rasburicase autoimmune antibodies and anaphylaxis due to repeated exposure, rasburicase was not used during the fourth and fifth courses of MTX.

The course of treatment in the present study was complicated by a history of gout and hyperuricemia, which were well-controlled without medication immediately before chemotherapy. There are only a few reports in the literature on TLS associated with gout, although patients with preexisting elevated uric acid levels due to gout prior to the diagnosis of malignancy are mentioned in the TLS risk classification [7]. One report described a patient with a history of gout who developed TLS after a treatment with imatinib for a metastatic gastrointestinal stromal tumor. The treatment was complicated by hyperkalemia, and the patient died 11 days after the initial treatment with imatinib [12]. In another report, a patient with no history of gout or hyperuricemia suffered from TLS and gouty arthritis of the knee after radiofrequency ablation for a hepatocellular carcinoma [13].

Schlafen11 (SLFN11) is a DNA/RNA helicase whose expression correlates well with the sensitivity to DNA-damaging agents [14]. The present case showed a relatively high expression of SLFN11 by immunohistochemistry, suggesting sensitivity to chemotherapy (Figure 2C). Interestingly, a recent report suggested that nongerminal center B (non-GCB)-type lymphomas, the predominant type in PCNSLs, have higher SLFN11 levels than the GCB-type ones [15], although the SLFN11 expression in PCNSLs has not been studied to date. We analyzed the CellMinerCDB database and found a weak positive correlation between the SLFN11 expression and the sensitivity to methotrexate in lymphoma cell lines (r = 0.44, Appendix A) [16]. Thus, SLFN11 is a candidate for predicting sensitivity to chemotherapy, thus anticipating the risk of TLS and acute uric acid nephropathy.

In the present case, acute uric acid nephropathy was observed after the first course of MTX. Fortunately, the renal function normalized, induction chemotherapy was successfully completed, and post-contrast MR images obtained after the induction therapy showed a CR. We believe that in the present case, acute uric acid was a sign of a good response to chemotherapy; thus, continuation of chemotherapy was actively pursued.

We would like to emphasize that, given the patient’s history of hyperuricemia and gout and large tumor volume, screening and prevention of side effects due to TLS, including rasburicase administration before the initiation of chemotherapy, should have been considered as salvage therapy after the onset of ARF could severely impact the treatment schedule, regimen, and outcome [17].

The limitations of this study include the fact that this was a case report, and definite conclusions cannot be made based on this single report. Considering the history of gout and hyperuricemia, although serum uric acid levels were not high before the initiation of chemotherapy, more aggressive measures should have been taken to prevent acute uric acid nephropathy. However, given the paucity of reports on acute uric acid nephropathy and TLS during the treatment of PCNSLs, a large-scale prospective study investigating the effects of serum uric acid and the renal function after MTX administration in PCNSLs is warranted.

## 4. Conclusions

PCNSLs are chemosensitive tumors, and routine screening of serum uric acid and metabolites should be performed during the induction chemotherapy, especially when the initial tumor volume is large. When serum uric acid levels are high, rasburicase, febuxostat, and/or allopurinol should be considered to prevent acute uric acid nephropathy.

## Figures and Tables

**Figure 1 jcm-11-05548-f001:**
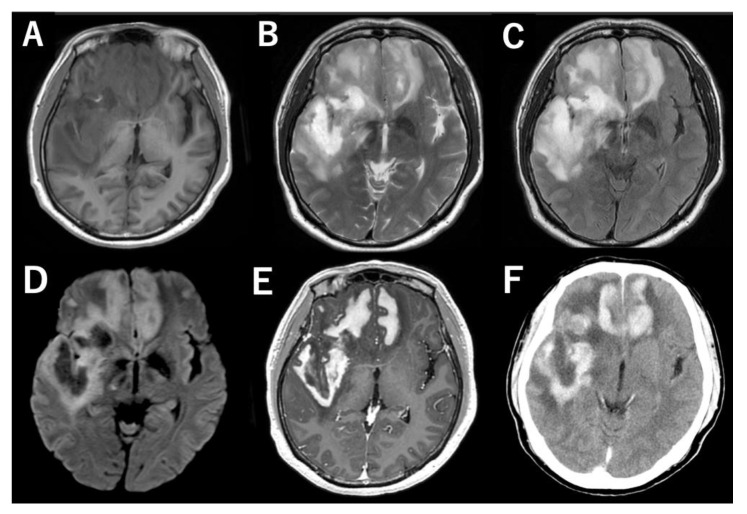
MR and CT images show a large homogenously enhancing lesion spanning the bilateral frontal lobes and the right temporal lobe. (**A**) T1-weighted image (T1WI), (**B**) T2WI, (**C**) FLAIR, (**D**) DWI, (**E**) post-contrast MRI, (**F**) post-contrast CT.

**Figure 2 jcm-11-05548-f002:**
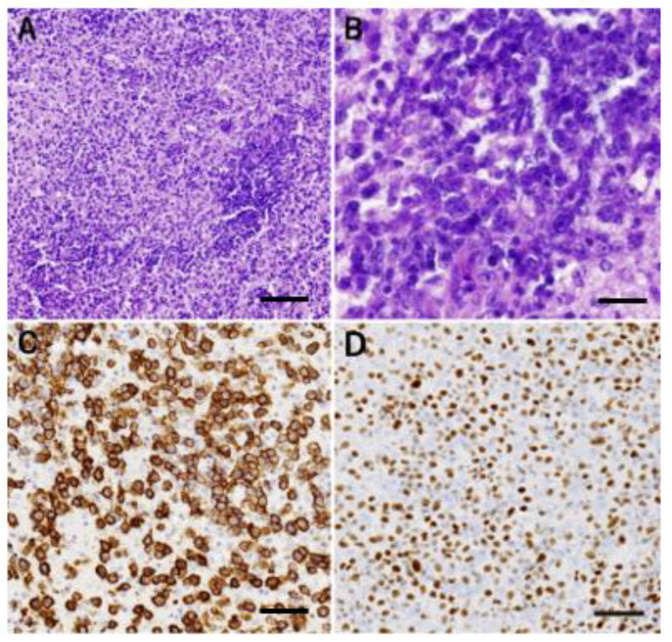
(**A**,**B**) Histopathological examination revealed perivascular and diffuse distribution of relatively large lymphoma cells and smaller reactive lymphocytes (hematoxylin–eosin staining). (**C**) The large cells were positive for CD20. (**D**) The tumor cells were highly positive for SLFN11, suggesting sensitivity to chemotherapy. Scale bar = 80 μm (**A**), 20 μm (**B**), 40 μm (**C**,**D**).

**Figure 3 jcm-11-05548-f003:**
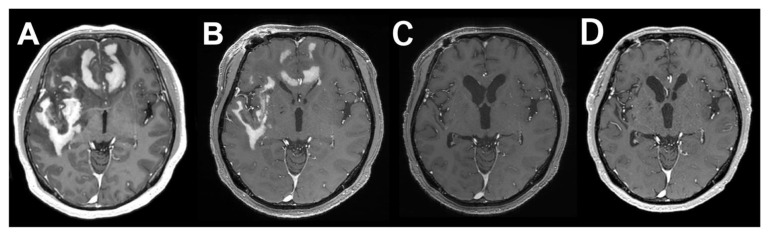
Post-contrast MR images showing **a** dramatic response to steroids and chemotherapy. (**A**) At presentation. (**B**) After steroid treatment. (**C**) After three cycles of methotrexate (MTX). (**D**) After five cycles of MTX.

**Figure 4 jcm-11-05548-f004:**
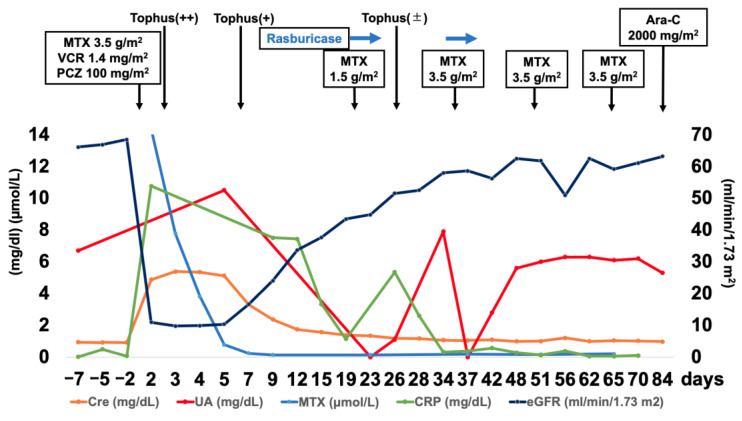
Timeline of the treatment, symptoms, and blood test results showing an elevation of uric acid and CRP (possibly reflecting the activeness of gout attack) and acute renal failure after the first administration of MTX. Bouts of tophus and uric acid elevation were subsequently controlled with rasburicase, and induction chemotherapy was ultimately completed. Abbreviations: Ara-C, cytarabine; Cre, creatine; CRP, C-reactive protein; eGFR, estimated glomerular filtration rate; MTX, methotrexate; PCZ, procarbazine; UA, uric acid; VCR, vincristine.

## Data Availability

Not applicable.

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
