# Peer review of "Successful Treatment of Acute Uric Acid Nephropathy with Rasburicase in a Primary Central Nervous System Lymphoma Patient Showing a Dramatic Response to Methotrexate—Case Report"

_jcm, 2022, doi:10.3390/jcm11195548_

Round 1
Reviewer 1 Report
In this manuscript, the authors report on a case of extensive PCNSL successfully treated with chemotherapy (MPV regimen) showing complete response in follow-up imaging. They importantly outline treatment strategies for tumor lysis syndrome as a chemotherapy-related side effect. In detail, they describe the use of rasburicase on lowering elevated uric acid blood-levels with clinically accompanying acute renal failure and gout attack. Pre- and post-treatment MRI imaging is shown as well as neuropathological examination. Here, the authors discuss the use of SLFN11-expression levels in tumors for prediction of chemotherapy sensitivity.
Overall, the paper is well written: The introduction and the discussion clearly present the background and impact as well as limitations. The figures facilitate the understanding of the manuscript. The case report is well structured.
In my opinion figure 4 could be markedly improved by: a) including a more detailed figure legend per se, b) including explanations of the used abbreviations, c) stating appropriate units for x-axis (I assume days from the text?) and y-axis (which lab parameter belongs to which y-axis scale – mg/dl left? ml/min/1.73m2 right?). In figure 2, factor of magnification from A to B (x4?) could be added for better visualization.
I feel that some in some passages, the use of the English language could be improved, or passages can be paraphrased (although not essential for understanding the scope of the manuscript), i.e.:
Line 28 “Since a good response to chemotherapy was observed, the latter was assumed” – A good response to chemotherapy does not rule out aforementioned MTX toxicity or speaks in favour for nephropathy. In my understanding the message was: Since MTX showed dramatic effect, treatment was continued despite acute renal failure - most probably - due to tumor lysis syndrome, which was treated parallelly.
Line 48/49 “memory loss from 1 month” please clarify or paraphrase – memory loss beginning one month prior to initial presentation or retrograde amnesia for events older than one month?
Line 108 “dramatic response (CRu)” – please introduce abbreviation CRu ïƒ unconfirmed complete response?
Line 121-125 Factors listed can be divided into two categories and attributed to either tumor-related risk factors (size, chemotherapy sensitivity, etc.) or patient/history-related risk factors (hx of gout/hyperuricemia, renal disease, etc.).
I would encourage the authors to elaborate on the importance of screening and prevention of side-effects due to tumor lysis syndrome, i.e. pre-chemotherapy treatment with rasburicase versus salvage therapy with rasburicase, etc. after onset of acute kidney failure.
Author Response
Dear editor,
We thank the reviewers for a thorough review of our paper, and generally constructive comments. We have now revised the paper based on suggestions from the 2 reviewers. We hope that with these improvements, the paper will now be suitable for publication in JCM.
Reviewer 1
In this manuscript, the authors report on a case of extensive PCNSL successfully treated with chemotherapy (MPV regimen) showing complete response in follow-up imaging. They importantly outline treatment strategies for tumor lysis syndrome as a chemotherapy-related side effect. In detail, they describe the use of rasburicase on lowering elevated uric acid blood-levels with clinically accompanying acute renal failure and gout attack. Pre- and post-treatment MRI imaging is shown as well as neuropathological examination. Here, the authors discuss the use of SLFN11-expression levels in tumors for prediction of chemotherapy sensitivity.
Overall, the paper is well written: The introduction and the discussion clearly present the background and impact as well as limitations. The figures facilitate the understanding of the manuscript. The case report is well structured.
In my opinion figure 4 could be markedly improved by: a) including a more detailed figure legend per se, b) including explanations of the used abbreviations, c) stating appropriate units for x-axis (I assume days from the text?) and y-axis (which lab parameter belongs to which y-axis scale – mg/dl left? ml/min/1.73m2 right?). In figure 2, factor of magnification from A to B (x4?) could be added for better visualization.
We thank the reviewer for a very help comment. We have now changed Figure 4 as suggested to make it easier for the reader to understand. The factor of magnification from A to B is indeed x4 as mentioned, which can be calculated from the scale we have provided (The scale bar is 80 µm in A as opposed to 20 µm in B.).
I feel that some in some passages, the use of the English language could be improved, or passages can be paraphrased (although not essential for understanding the scope of the manuscript), i.e.:
We thank the reviewer for the comment. In the interest of the short, allotted time before resubmission, we have used R Pubsure to check for any grammatical or word usage errors in the manuscript and have uploaded the certificate.
Line 28 “Since a good response to chemotherapy was observed, the latter was assumed” – A good response to chemotherapy does not rule out aforementioned MTX toxicity or speaks in favour for nephropathy. In my understanding the message was: Since MTX showed dramatic effect, treatment was continued despite acute renal failure - most probably - due to tumor lysis syndrome, which was treated parallelly.
We thank the reviewer for the important comment. We agree with the reviewer that a good response to chemotherapy does not rule out nephrotoxicity due to MTX. We have changed the sentence as suggested, to clarify the meaning. However, we decided against using the term “tumor lysis syndrome”, because this case did not meet the specific criteria.
Line 48/49 “memory loss from 1 month” please clarify or paraphrase – memory loss beginning one month prior to initial presentation or retrograde amnesia for events older than one month?
We have now clarified that memory loss began one month prior to initial presentation.
Line 108 “dramatic response (CRu)” – please introduce abbreviation CRu à unconfirmed complete response?
We have introduced the abbreviation CRu as suggested.
Line 121-125 Factors listed can be divided into two categories and attributed to either tumor-related risk factors (size, chemotherapy sensitivity, etc.) or patient/history-related risk factors (hx of gout/hyperuricemia, renal disease, etc.).
We have now divided risk factors for TLS in tumor-related risk factors and patient/history-related risk factors as suggested (lines 194-199) .
I would encourage the authors to elaborate on the importance of screening and prevention of side-effects due to tumor lysis syndrome, i.e. pre-chemotherapy treatment with rasburicase versus salvage therapy with rasburicase, etc. after onset of acute kidney failure.
We thank the reviewer for bringing up a very important point. TLS or acute uric acid nephropathy has not been extensively reported in PCNSL treatment, so we were unable to prevent it in the present case, but we agree with the reviewer that prevention is vital, so physicians need to be aware of the possibility of TLS or acute uric acid nephropathy during PCNSL treatment, especially when risk factors are present. We have now stressed this important point in the Discussion section (lines 256-259).
Reviewer 2 Report
I am glad to have an opportunity to evaluate this case report entitled “Successful treatment of acute uric acid nephropathy with rasburicase in a primary central nervous system lymphoma patient 3 showing dramatic response to methotrexate”. The authors here described a case with PCNSL experiencing uric acid nephropathy and treated successfully with rasburicase. The paper is well written, however there are some points that need to be addressed.
-
I believe there are so many authors in this case. Do they really contribute enough to be written as authors or some of them can be placed in the acknowledgment part. The authors should reconsider this issue.
-
In figure 3, I believe there is a typo, in the footnotes “(E)” should be “(D)”.
-
The author may use scales such as Naranjo scale to present the Adverse Drug Reaction Probability.
-
The authors may mention gout and TLS co-existence literature and unique parts of the case.
-
Were there any neurological assessments made before and after therapy such as mini mental test etc.
-
The patient has a history of gout, do the authors take any precaution of hyperuricemia before the chemotherapy started? Allopurinol prophylaxis, hydration etc.
-
Is there any reason for authors to initiate febuxostat not allopurinol after chemotherapy finished?
-
In the discussion, authors TLS risk groups and approaches can be more comprehensive.
-
The authors should mention limitations of the case report
-
In the conclusion the authors mentioned “For newly diagnosed PCNSL patients with large tumors or hyperuricemia, upfront usage of rasburicase should be considered to prevent it.” . However in the case report the authors presented rasburicase application in the acute urate nephropathy setting. I believe this conclusion is not correlated with the case.
Author Response
Reviewer 2
I am glad to have an opportunity to evaluate this case report entitled “Successful treatment of acute uric acid nephropathy with rasburicase in a primary central nervous system lymphoma patient 3 showing dramatic response to methotrexate”. The authors here described a case with PCNSL experiencing uric acid nephropathy and treated successfully with rasburicase. The paper is well written, however there are some points that need to be addressed.
- I believe there are so many authors in this case. Do they really contribute enough to be written as authors or some of them can be placed in the acknowledgment part. The authors should reconsider this issue.
We understand the concern of the reviewer, as there are many co-authors considering that this is a case report. Please understand that many specialists, including pharmacists, pathologists, neurosurgeons and oncologists were involved in treatment and work-up of this case, which included assessment of SLFN11 expression for prediction of chemosensitivity.
- In figure 3, I believe there is a typo, in the footnotes “(E)” should be “(D)”.
We thank the reviewer for pointing out this error. We have changed (E) to (D).
- The author may use scales such as Naranjo scale to present the Adverse Drug Reaction Probability.
We have now added the Naranjo scale to define the adverse drug reaction probability as “definite” (lines 170-172).
- The authors may mention gout and TLS co-existence literature and unique parts of the case.
We did an extensive literature search of reports of co-existence of gout and TLS but were unable to find many reports. One report described a patient with a past history of gout, who developed TLS after treatment of a metastatic GIST with imatinib. The patient died 11 days after initial treatment with imatinib and an autopsy was performed (Ref #12). In another report, a patient with no past history of gout or hyperuricemia, suffered from TLS and gouty arthritis of the knee after radiofrequency ablation for hepatocellular carcinoma (Ref #13). We have mentioned these two paper in the discussion section (lines 231-240).
- Were there any neurological assessments made before and after therapy such as mini mental test etc.
Neurological assessments, including Mini Mental State Examination and WAIS-IV, were performed before and after chemotherapy, showing a marked improvement. We have now added this data to the paper (lines 177-181).
- The patient has a history of gout, do the authors take any precaution of hyperuricemia before the chemotherapy started? Allopurinol prophylaxis, hydration etc.
We have not been able to find any reports of TLS or acute uric acid nephropathy after PCNSL treatment. Therefore, we neurosurgeons did not have the mind that acute renal failure due to tumor lysis may occur. We do routinely check for serum uric acid before initiation of chemotherapy. Hydration was performed before treatment but not allopurinol prophylaxis.
- Is there any reason for authors to initiate febuxostat not allopurinol after chemotherapy finished?
We understand that first line treatment for prevention of tumor lysis syndrome is allopurinol. We decided to use febuxostat instead of allopurinol because relapse of renal dysfunction was a concern after continuation of MTX. We have now added a meta-analysis showing that febuxostat is not inferior to allopurinol in TLS prophylaxis (Ref #11).
- In the discussion, authors TLS risk groups and approaches can be more comprehensive.
We have now discussed the risk groups and prophylaxis of TLS in diffuse large B cell lymphomas as suggested by the expert TLS panel (Reference #7, lines 199-200). We would like to stress that reports of TLS in PCNSL are very rare and is not specifically mentioned in the recommendations.
- The authors should mention limitations of the case report.
We have now added a paragraph at the end of discussions, listing the limitations of this case report, including that definitive conclusions cannot be drawn from this single case report, that more aggressive prophylaxis of TLS and acute uric acid should have been taken, and that large scale, prospective studies are awaited (lines 261-268).
- In the conclusion the authors mentioned “For newly diagnosed PCNSL patients with large tumors or hyperuricemia, upfront usage of rasburicase should be considered to prevent it.”. However, in the case report the authors presented rasburicase application in the acute urate nephropathy setting. I believe this conclusion is not correlated with the case.
We thank the reviewer for a very valid point. We have now changed the conclusion of the Abstract to, “Acute uric acid nephropathy should be considered when ARF occurs after initiation of MTX in PCNSL, especially in newly diagnosed PCNSL patients with large tumors or hyperuricemia.”